# Characterization of Base Oil Effects on Aged Asphalt Binders Considering Bicycle Road

**DOI:** 10.3390/ma16020624

**Published:** 2023-01-09

**Authors:** Jihyeon Yun, Hyun Joon Choi, Il-Ho Na, Hyun Hwan Kim

**Affiliations:** 1Materials Science, Engineering and Commercialization, Texas State University, San Marcos, TX 78666, USA; 2Insung A & T, Asan-si 31582, Republic of Korea; 3Korea Petroleum, Seoul 04427, Republic of Korea; 4Department of Engineering Technology, Texas State University, San Marcos, TX 78666, USA

**Keywords:** reclaimed asphalt pavement (RAP), rolling thin-film oven (RTFO), pressure aging vessel (PAV), base oil, asphaltene

## Abstract

Demand for various bicycles and sharing systems has constantly been growing worldwide as they improve the quality of life and promote eco-friendly transportation. Accordingly, it is inevitable that bicycle roads should be expanded. As bicycle roads have a relatively lower load applied than automobile roads, adopting a design method that uses a high reclaimed asphalt pavement (RAP) content can be beneficial. However, much uncertainty still exists about the relation between the mixing method and application in field sites, without appropriately considering the quality control of the rejuvenator. Therefore, this study aims to demonstrate the effect of base oil as a rejuvenator on aged binders, considering the use of a high RAP content for bicycle roads. To prepare the aged binder, a rolling thin-film oven (RTFO) and pressure aging vessel (PAV) were used to imitate the life cycle of asphalt pavement from production to service life, and then three contents of aged binder (0%, 50%, and 100%) were added and mixed with fresh PG 64-22 base binder. Finally, each type of prepared aged asphalt binder was blended with three different base oil contents (0%, 5%, and 10%). The results indicated that (1) the addition of base oil effectively decreased the viscosity of aged binders, (2) aged binders containing base oil showed less G*/sin *δ* compared to originally aged binders, and (3) the application of base oil improves the cracking properties of the aged binder by decreasing stiffness. In conclusion, the most striking observation from the data analysis from the Superpave test and statistical results was the effect of reducing the asphaltene portion based on the use of base oil in the aged binder. Therefore, using base oil in RAP can enable the application of a high RAP content to the bicycle road.

## 1. Introduction

Recently, as the quality of life has improved due to the growth in national income, the number of vehicles owned as a convenient means of transportation has increased continuously. However, the rapid increase in the use of automobiles has caused traffic congestion and an environmental problem. Accordingly, governments are trying to solve many issues, such as air pollution, noise, traffic congestion, and accidents created by vehicles. To this end, a policy is suggested to promote the use of bicycles. As a non-motorized, carbon-free means of transportation, bikes are in the spotlight as a sustainable transportation method with various advantages, such as solving environmental pollution and promoting public welfare [1].

Furthermore, as the demand for general and electric bicycles increases [2] and bicycle-sharing systems increase rapidly worldwide [3], it is necessary to expand the bicycle road, considering materials to ensure the durability of bicycle roads. In the context of these materials, diverse asphalt mixtures can be valuable pavement materials for designing bicycle roads [2,4].

In general, as bicycle roads have a relatively lower load applied than automobile roads, it can be beneficial to adopt a design method for paving the road with a limited budget and utilizing more reusable resources than automobile roads require. For instance, considering that the amount of reclaimed asphalt pavement (RAP) used usually is less than 25% in the case of general asphalt pavement affected by vehicle load [5], a relatively high RAP content can be applied to bicycle roads. However, the current study of bicycle roads has failed to consider the higher RAP content due to the low stripping resistance [2]. Therefore, it is necessary to solve the adverse effects such as the high stiffness and low viscoelasticity of the aged asphalt binder.

Recently, various methods to alleviate the disadvantages of using RAP have been developed using a softer virgin binder and adopting warm mix asphalt (WMA) technology, including using multiple types of oil [6,7,8]. This rejuvenator’s benefits have motivated researchers to investigate further mixtures with increasing RAP content [9]. However, even though a rejuvenator is applied to utilize RAP, increasing RAP content by more than 20% reduces the ability of fatigue and thermal cracking, which are significant obstacles to maximizing RAP utilization [10,11]. Although some research has been carried out using 100% RAP, much uncertainty still exists about the relation between the mixing method and application in field sites, without appropriately considering the quality control of the rejuvenator [12,13]. In addition, far too little attention has been paid to the sources of rejuvenating agents, such as various oils, which are very diverse, and their management is also unclear. Therefore, reviewing a more effective rejuvenator for stable quality control and more detailed analysis is necessary to utilize 100% RAP. In the context of stable and high-quality oil for use as a rejuvenator, base oil can be considered since this oil is reinvented as a premium oil with various purposes, such as gasoline engine oils and diesel engine oils. Due to this particular use of base oil to create a more developed product, managing the quality is necessary. Hence, base oil can be considered to achieve a stable quality asphalt mixture using 100% RAP.

In this study, the main objective was to assess the effect of base oil on the aged binder. First, to extract the aged binder, a rolling thin-film oven (RTFO) and pressure aging vessel (PAV) were used to imitate the life cycle of asphalt pavement from production to service life, and then each content of aged binder (0%, 50%, and 100%) was added to PG 64-22. Next, each type of aged asphalt binder prepared was blended with a base oil (0%, 5%, and 10%). Finally, as an evaluation method for the asphalt binder, the Superpave test was adopted to understand how the base oil behaved in each aged asphalt binder. Figure 1 shows the flow chart used for this study.

## 2. Experimental Design

### 2.1. Materials

In this study, the base asphalt binder (PG 64-22) whose properties are shown in Table 1 was utilized to prepare aged asphalt binders and modify the asphalt binder with base oil, as shown in Figure 2. One of the purposes of this paper is to evaluate the effect of base oil on aged asphalt binder considering RAP. PG 64-22 is the common binder for asphalt pavement. This means that it produces much RAP and can verify the application of base oil on PG 64-22-based RAP. This is why PG 64-22 was selected for this study. Base oil is a transparent viscous liquid applicable for various purposes, such as vehicle engine oil and lubricant for mechanical equipment. To confirm the optimized content of base oil, amounts of 5% and 10% were applied. The properties of the base oil considered in this study are presented in Table 2.

### 2.2. Production of Asphalt Binders

To evaluate how asphalt binder behaves at different temperatures on the Superpave test, the first step was to prepare aged asphalt binder based on the aging procedure of the rolling thin-film oven (RTFO) and pressure aging vessel (PAV) to imitate a binder derived from reclaimed asphalt pavement (RAP). The aged binder was mixed with PG 64-22 binder, as shown below.

Binder A: PG 64-22(100%) + Aged binder (0%).Binder B: PG 64-22(50%) + Aged binder (50%).Binder C: PG 64-22(0%) + Aged binder (100%).

Each type of prepared asphalt binder was blended, adding 0%, 5%, and 10% of base oil by using a low-shear mixer (300 rpm) at a temperature of 160 °C for 10 min.

### 2.3. Superpave Binder Tests

Superpave binder tests were adopted to evaluate the characteristics of asphalt binders. To conduct each binder test, all asphalt binders were conditioned in three stages of original, short-term aging (RTFO), and long-term aging (PAV) conditions. Rotational viscosity (Test Method D 4402), dynamic shear rheometer (Test Method D7175), and bending beam rheometer (AASHTOT 313) tests were performed. Figure 3 shows the procedure of the Superpave binder test in this study.

The viscosity of asphalt binder is a key to evaluating whether its workability can be adequate on the field site. To evaluate the workability, asphalt binder was assessed by a Brookfield rotational viscometer (Figure 4) at a temperature of 135 °C, applying the 27 cylindrical spindle and constant 20 rpm with 10.5 g of original asphalt binder in accordance with AASHTO T 316. The operation time to obtain the viscosity results was adopted as 20 min for all samples. The rheological property using a dynamic shear rheometer (DSR) was analyzed with samples of unaged, RTFO, and RTFO + PAV binder according to ASTM D7175. The results of G*/sin *δ* were collected on the basis of the results of the complex shear modulus (G*) and sine (*δ)* of the phase angle at 64 °C. To figure out the fatigue cracking at ambient and low temperatures, RTFO + PAV binders were used to obtain the results of G*·sin *δ* by DSR and of stiffness and *m*-value by a bending beam rheometer (BBR). In the case of G*·sin *δ*, it was measured at 25 °C to examine fatigue cracking properties. For cracking properties at low temperatures, the BBR test began by forming asphalt beams (125 × 6.35 × 12.7 mm) and placing these beams inside of the BBR based on the suggested standard. Finally, the stiffness and *m*-value were calculated by applying a constant load of 100 g to the center of the asphalt beam. At this point, in order to analyze the comprehensive effect of base oil on aged asphalt binder, considering that the stiffness increase when RAP is applied to the asphalt mixture, diverse temperatures of −12 °C, −18 °C, and −24 °C were considered.

### 2.4. Statistical Evaluation

Statistical significance was analyzed through the use of the Statistical Package for the Social Sciences (SPSS) program to perform the analysis of variance (ANOVA) based on a Fisher’s Least Significant Difference (LSD) comparison with α = 0.05. The ANOVA was conducted to identify whether a significant difference appear between means for samples. In accordance with the results of the ANOVA, multiple means were compared simultaneously by comparing the difference between the two paired means with the LSD value to review whether the difference between the means was significant.

## 3. Results and Discussion

### 3.1. Rotational Viscosity

Rotational viscosity (RV) is used to predict the workability between periods of production and compaction for an asphalt mixture. For instance, when producing asphalt mixtures with high or low viscosity, the asphalt mixture can be undesirably compacted, making it challenging to obtain optimal density. To avoid these problems, the RV standard is generally suggested below 3000 cP at 135 °C. In this study, RV was evaluated at the temperature of 135 °C to determine whether base oil can affect the aged asphalt binder in terms of viscosity. The RV result for each binder is shown in Figure 4. Overall, as the aged asphalt binder increased, the RV also increased.

On the other hand, as expected, the decreasing trend for RV was witnessed with the addition of base oil. These results show increased viscosity with asphaltene caused by aged asphalt binder. In contrast, the viscosity was reduced due to the regenerative effect due to the addition of base oil. The RV result of using only PG 64-22 as a base binder showed the lowest value from about 500 cP to 200 cP by adding base oil from 0% to 10%. In the case of using 50% aged binder, the RV results were higher than using PG 64-22 binder, showing a decreasing trend from about 800 cP to 300 cP in each content of 0% and 10% base oil. As expected, the RV results from using 100% aged binders saw the highest value of around 1300 cP in 0% base oil. Because the base binder was aged, the asphaltene increased and became stiffer [14]. However, the viscosity of 100% aged binder dropped constantly to a low of approximately 500 cP in 10% base oil. The most notable finding of this experiment was that even though 100% aged binder was utilized as a base binder, using 10% base oil made it possible to rejuvenate the aged binder as much as the original PG 64-22 binder. It can be seen that base oil is effective on the aged binder, decreasing the high viscosity caused by asphaltene.

The RV results were assessed by adopting statistical significance (α = 0.05) based on the ANOVA. The result is shown in Table 3. The overall tendency for this analysis among all binders revealed statistical significance, meaning that the aged binder and base oil can affect binder properties. However, no significant difference between PG 64-22 (100%) and the binder containing 50% aged binder with 5% base oil was found. These results indicate that using the base oil can positively affect the aged binder, rejuvenating itself as much as PG 64-22.

### 3.2. Rutting Properties

#### 3.2.1. G*/sin δ at Original State

DSR is the most common tool for assessing the rheological property of asphalt binder based on the G*/sin *δ* factor. For example, the higher the G*/sin *δ* that the asphalt binder shows, the higher the indicated rutting resistance. However, if the asphalt binder is excessively stiff, even if the G*/sin *δ* is high, there will be a risk of cracking, or workability cannot be ensured. Therefore, additional evaluations should be considered to look into the overall properties of asphalt binders. In this study, the original G*/sin *δ* was measured at 64 °C, as shown in Figure 5. A dramatic increase in G*/sin *δ* was seen from 0% to 100% aged binder, with the value reaching a peak of approximately 15 kPa without using the base oil. This is because the stiff characteristics originated from the proportion of asphaltene increasing during the aging procedure. According to several studies [15,16], the asphaltene portion in asphalt binder increases as it is aged by oxidation. However, using 5% base oil, the value of G*/sin *δ* dropped to almost half from using 0% base oil in each binder. In the case of using 10% base oil, the value decreased by about four times or more. Therefore, the base oil is considered adequate for the aged binder because the asphaltene is well diminished by softening the asphalt binder. Another implication of these results found in this study is the potential that even if 50% of the aged asphalt binder is used, it can be regenerated as much as PG 64-22 binder using 5% base oil.

For statistical analysis using ANOVA, the statistical significance of the G*/sin *δ* value for the original condition was evaluated (Table 4). This investigation showed that there was significance for all binders, which means that the aged binder and the base oil influence each other. Moreover, this statistical result corresponds with the bar chart for the original G*/sin *δ.*

#### 3.2.2. G*/sin δ after Short-Term Aging (RTFO Process)

RTFO is a procedure for simulating short-term aging in the field condition between the production and delivery of asphalt mixture. Therefore, G*/sin *δ* for the RTFO condition was evaluated to consider this field condition, as shown in Figure 6. As with the results of the original condition, the increasing trend for the G*/sin *δ* value was seen as the binder aged, while a decreasing trend for it was witnessed by using the base oil. The result in an aged binder without using base oil saw the most significant change, with a more than seven times increase from around 4 kPa to 30 kPa. G*/sin *δ* using 5% base oil dropped to almost half of the percentage from the value of using 0% base oil. Concerning the 10% base oil results, there was a significant decreasing trend for the G*/sin *δ*. Like the results in the original condition, when 5% base oil was utilized in a binder using 50% aged binder, it was found to show the characteristics of PG 64-22. This result indicated an understanding of how base oil behaves in the aged binder as a rejuvenator, as mentioned earlier in the results of the original condition.

The statistical significance of the change in G*/sin *δ* for the RTFO condition was evaluated in the same way as the original condition, comparing each binder. As Table 5 shows, a significant difference was measured among all binders. This result means that binders containing a high proportion of asphaltene behaved differently compared to PG 64-22. Meanwhile, the effect of using the base oil for the aged binder was also shown based on the distinct significant difference in the statistical analysis.

### 3.3. Fatigue Cracking Property

To ensure the resistance of fatigue cracking, the lower G*·sin *δ* was considered a factor to improve this resistance, suggesting that the value should be less than 5000 kPa. If the asphalt binder is too stiff, the value will rise, meaning that asphalt pavement can suffer fatigue cracking. The G*·sin *δ* was measured using DSR at the temperature of 25 °C, and the results are shown in Figure 7. Overall, there was a clear increasing trend for G*·sin *δ* as the asphalt binder aged. On the other hand, using the base oil was shown to have a positive effect on decreasing the value. In particular, the G*·sin *δ* of 100% aged binder peaked at around 8000 kPa from 4000 kPa of PG 64-22. By contrast, using 10% base oil appeared to have a rejuvenating effect, resulting in a roughly twofold decrease from each aged binder. In this respect, even though 100% aged binder was used, even if 5% base oil was added to this aged binder, the aged binder was shown to regenerate as much as the PG 64-22 binder. This result shows that the asphaltene content, which makes the asphalt stiff, decreased, lowering G*·sin *δ* and rejuvenating the aged binder.

The statistical significance for the G*·sin *δ* value depending on the effect of using base oil on the aged binder was investigated. As Table 6 shows, the results revealed that the aged binder could have a significant effect on the G*·sin *δ* value by increasing the stiffness. Furthermore, the aged binder containing base oil was determined to have a significant difference among binders using base oil (0%, 5%, and 10%). In this statistical result using data for fatigue cracking as a factor, most of the data showed statistical significance among all binders, which means that the aged binder and base oil affect PG 64-22 properties. In particular, the base oil was confirmed to exhibit a regenerative effect for the aged binder.

### 3.4. Thermal Cracking Resistance at Low Temperature

In general, it is known that the aged binder has a risk of brittle fracture at low temperatures due to increased stiffness. Therefore, it is essential to consider this factor when using an aged binder, and as a standard, less than 300 MPa in stiffness and more than 0.3 *m*-value are suggested. In this study, to evaluate the fatigue cracking resistance at such low temperatures, the BBR test was conducted at comprehensive temperatures of −12 °C and −18 °C, including the extreme temperature of −24 °C.

As a result of the test conducted at −12 °C (Figure 8), the trend of increasing the stiffness and decreasing the *m*-value was witnessed when the asphalt binder was aged. Up to 100% aged binder being used, the stiffness increased marginally, reaching around 260 MPa. On the other hand, the *m*-value showed a decreasing trend. The base oil significantly affected each binder using an aged binder (0%, 50%, and 100%) by reducing the value more than twofold by adding 5% base oil. However, the results did not appear to differentiate the properties of each aged binder distinctly, considering that the test temperature was relatively low to determine how the aged binder behaved. Nevertheless, the base oil was shown to effectively lower the stiffness of each binder, which was considered to be decomposed into asphaltene.

In the results conducted at a temperature of −18 °C shown in Figure 9, the stiffness exceeded 300 MPa without using the base oil. However, all data showed that the stiffness was achieved using a base oil resulting in less than 300 MPa. It was shown that the stiffness decreased to about 200 MPa at each aged binder containing 5% base oil, but the result of using 100% aged binder revealed that the *m*-value was not over 0.3. Even with this result, the base oil was considered to have a positive effect due to the dramatic decrease in stiffness and reducing asphaltene in the aged binder.

In this study, the extreme temperature of −24 °C was considered to comprehensively evaluate the effect of base oil on the aged binder. As Figure 10 shows, and as expected, the stiffness rose sharply to a peak of almost 800 MPa with 100% aged binder. In addition, even though 5% base oil was added to each binder, the stiffness remained high between around 400 MPa and 500 MPa. However, using 10% base oil made it possible to reduce the stiffness to less than 300 MPa, resulting in around 200 MPa for each binder. This result showed that the base oil effectively decomposed asphaltenes, suggesting that aging binders can be used even at extremely low temperatures. In terms of *m*-value, there was a drop to under 0.3 in all data, but the dramatic increasing trend of *m*-value was seen using base oil compared to the results at −12 °C and −18 °C. Overall, this result confirmed that asphaltene made the binder body brittle at this extremely low temperature and that the base oil effectively relieved this high stiffness, as mentioned earlier, decreasing asphaltene.

The statistical significance of the change in the stiffness and *m*-value as a function of each aged binder was evaluated at temperatures of −12 °C, −18 °C, and −24 °C, as shown in Table 7, Table 8 and Table 9. Overall, the results indicated that aged binder and base oil play a significant role in changing the properties of stiffness and *m*-value when compared within each aged binder. This result revealed that base oil at each temperature affected the aged binder, lowering stiffness and increasing the *m*-value. In other words, the statistical analysis of the evaluation of fatigue cracking resistance at low temperatures means that the base oil has a positive effect of regenerating the aged binder while decomposing asphaltene. In addition, the *m*-value at the extremely low temperature of −24 °C, when the base oil was used for the 100% aged binder, was observed to regenerate as much as PG 64-22 asphalt binder, showing a non-significant value.

## 4. Summary and Conclusions

This study was conducted to see the probability of applying the RAP to an entire pavement to construct a bicycle road with relatively little traffic load. In this study, three types of asphalt binders—PG 64-22(100%) + Aged binder (0%), PG 64-22 (50%) + Aged binder (50%), and PG 64-22 (0%) + Aged binder (100%)—were prepared, and then each binder was blended with base oil to investigate the effect of base oil on the aged binder. A series of Superpave tests were performed to evaluate the performances of recycled binders. Based on the results, the following conclusions were drawn.

Adding base oil to aged asphalt binder showed a decreasing trend in rotational viscosity at 135 °C. The notable finding is that even though 100% aged binder was utilized as a base binder, using 10% base oil made it possible to rejuvenate the aged binder as much as the PG 64-22 binder. Therefore, it is considered that base oil is effective in decreasing the high viscosity caused by asphaltene in aged binders.

In the case of the rheological property based on G*/sin *δ* at 64 °C of the original and RTFO conditions, a considerable increase of G*/sin *δ* was observed from 0% to 100% aged binder due to the increased stiffness. However, the decreasing trend for G*/sin *δ* at both conditions was witnessed as base oil was added to the aged binders. Thus, the base oil is deemed to reduce the asphaltene for softening the aged binder considering the results of rutting properties.

From the fatigue cracking property measured at 25 °C, the base oil contents were shown to decrease the value positively. Even though 100% aged binder was used, even if 5% base oil was added to this aged binder, the aged binder was shown to regenerate as much as the PG 64-22 control binder. This result revealed that the asphaltene, which makes the asphalt stiff, is diminished, reducing G*·sin *δ* and rejuvenating the aged binder.

For the thermal cracking resistance at low temperatures of −12, −18, and −24 °C, the trend of increasing the stiffness and decreasing the *m*-value was observed when the asphalt binder was aged. The results at −12 °C did not appear to differentiate the properties of each aged binder distinctly due to the relatively high testing temperature. However, for the stiffness and *m*-value measured at −18 °C and −24 °C, it was confirmed that asphaltene made the binder body brittle at these temperatures and that the base oil effectively relieved this high stiffness, decomposing asphaltenes in the aged binder.

In summary, as a result of the Superpave test and statistical analysis, the asphaltene decomposition effect was demonstrated when the base oil was mixed with the aged asphalt binder. However, there are still many uncertainties regarding the utilization of base oils for direct field application. Therefore, in order to present a practical plan for using base oil in asphalt mixtures containing high RAP content, a comprehensive study, including an evaluation of RAP mixtures, is required.

## Figures and Tables

**Figure 1 materials-16-00624-f001:**
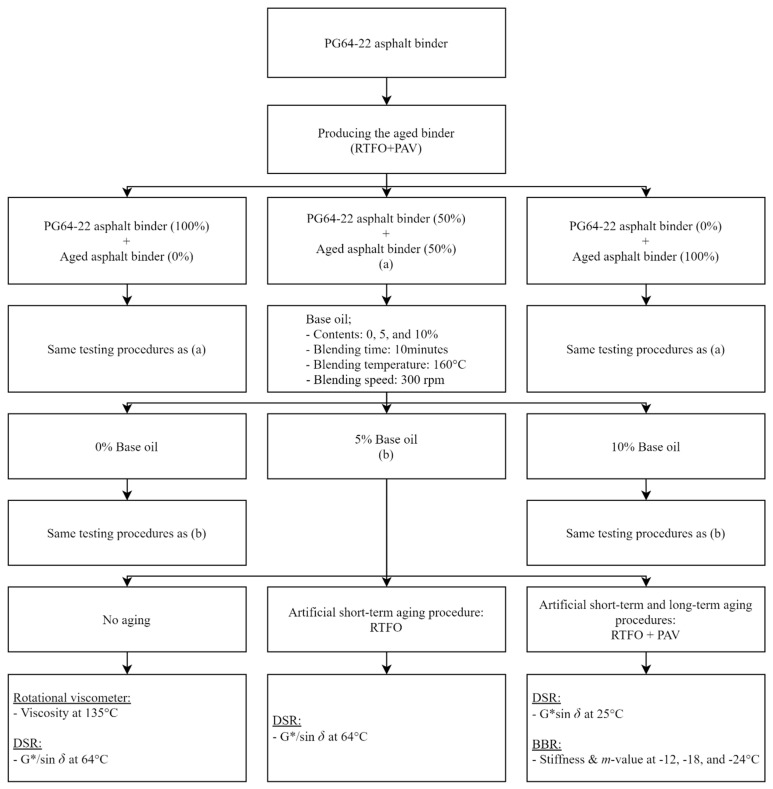
Flow chart of experimental design procedure in this study.

**Figure 2 materials-16-00624-f002:**
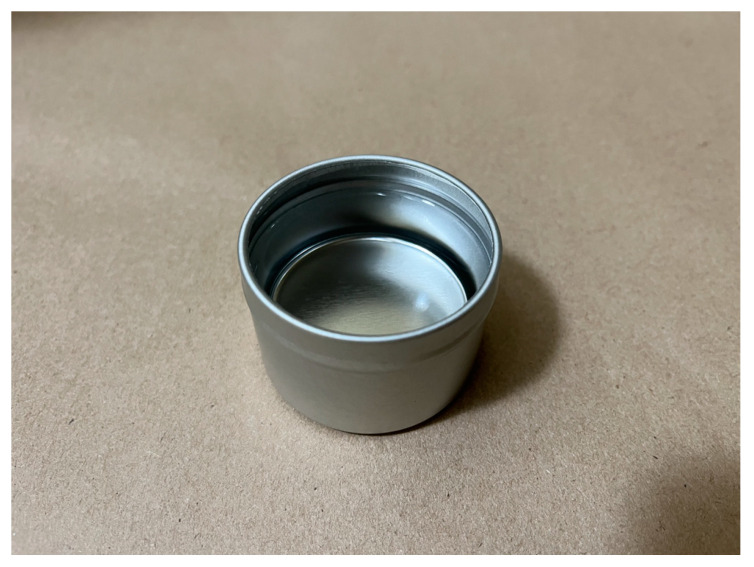
Base oil used in this study.

**Figure 3 materials-16-00624-f003:**
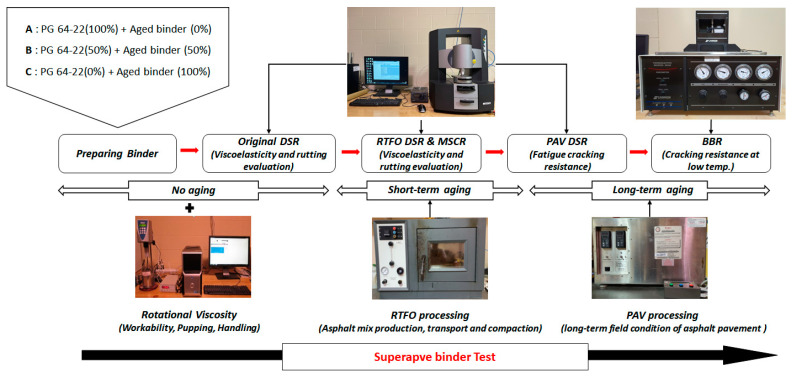
Superpave binder test procedure.

**Figure 4 materials-16-00624-f004:**
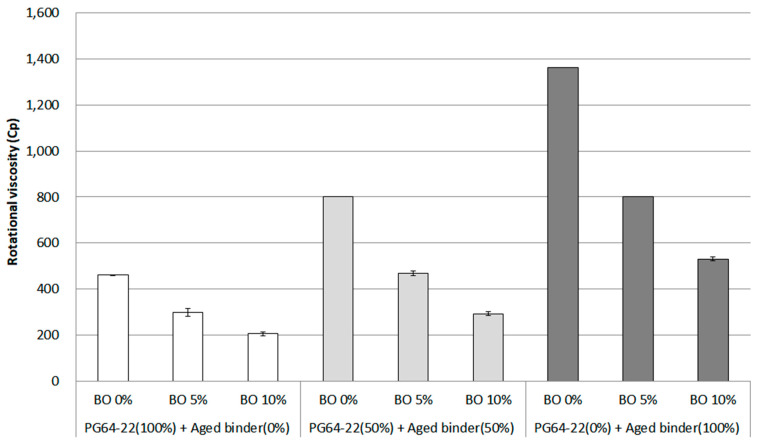
Rotational viscosity of each asphalt binder containing the base oil at 135 °C.

**Figure 5 materials-16-00624-f005:**
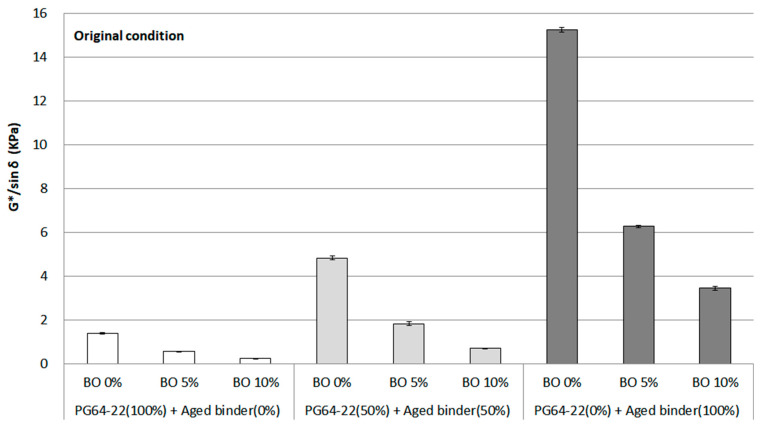
G*/sin *δ* of each asphalt binder containing the base oil for original condition.

**Figure 6 materials-16-00624-f006:**
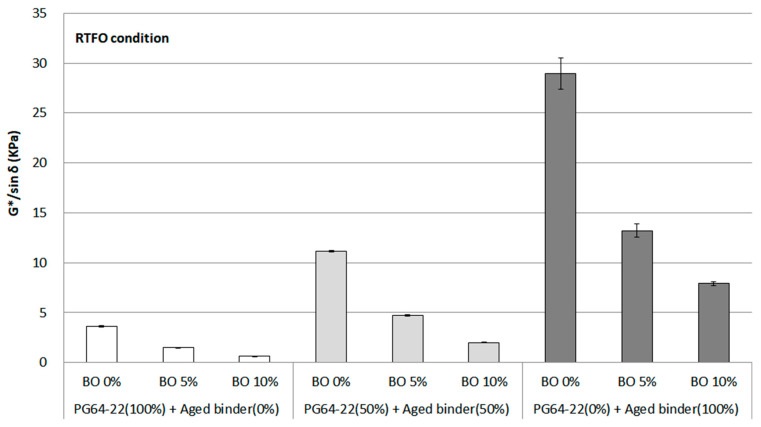
G*/sin *δ* of each asphalt binder containing the base oil for RTFO condition.

**Figure 7 materials-16-00624-f007:**
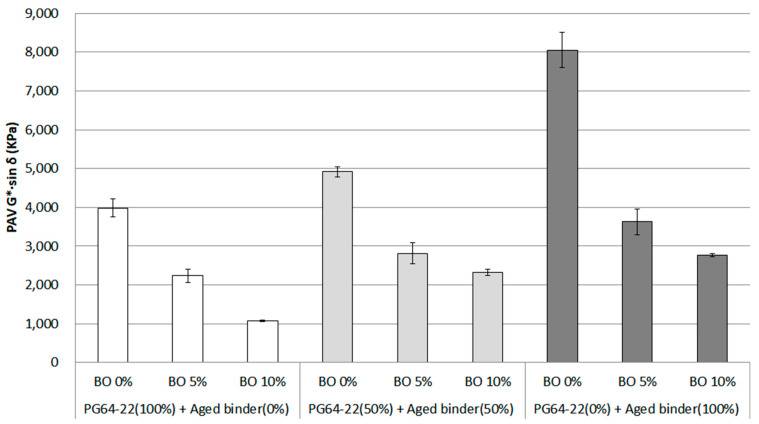
G*·sin *δ* of each asphalt binder containing the base oil for RTFO + PAV condition.

**Figure 8 materials-16-00624-f008:**
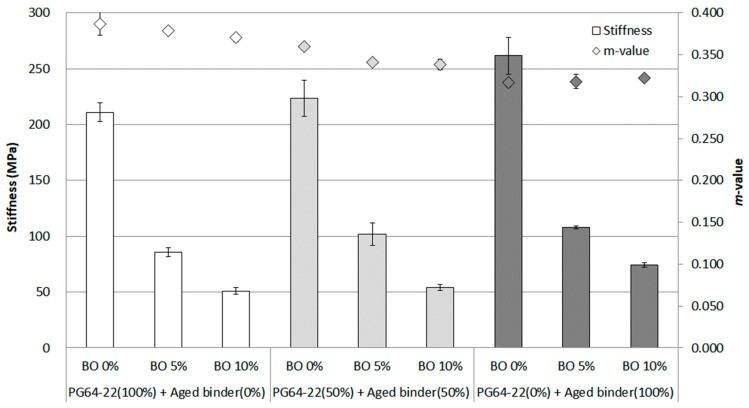
Stiffness and *m*-value of each asphalt binder containing the base oil for RTFO + PAV condition at −12 °C.

**Figure 9 materials-16-00624-f009:**
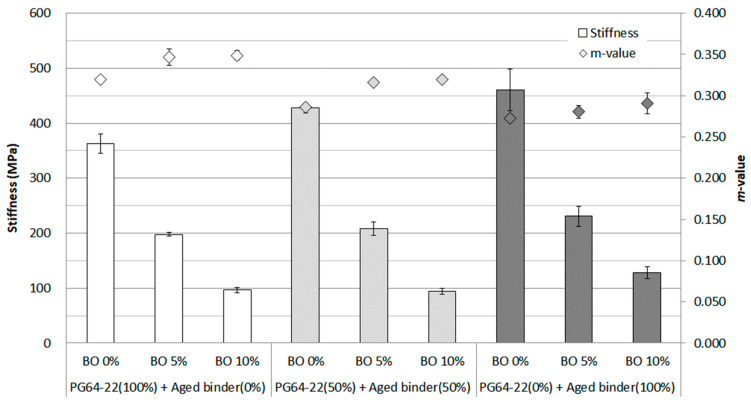
Stiffness and *m*-value of each asphalt binder containing the base oil for RTFO + PAV condition at −18 °C.

**Figure 10 materials-16-00624-f010:**
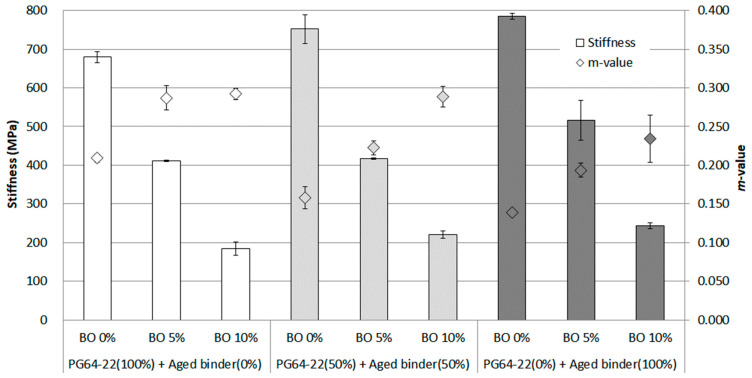
Stiffness and *m*-value of each asphalt binder containing the base oil for RTFO + PAV condition at −24 °C.

**Table 1 materials-16-00624-t001:** Properties of base asphalt binder (PG 64-22).

	Test Properties	Test Result
Original binder	Viscosity @ 135 °C (cP)	462
	G*/sin *δ* @ 64 °C (kPa)	1.38
RTFO aged binder	G*/sin *δ* @ 64 °C (kPa)	3.65
RTFO + PAV aged binder	G*·sin *δ* @ 25 °C (kPa)	3983
	Stiffness @ −12 °C (MPa)	211
	*m*-value @ −12 °C	0.387

**Table 2 materials-16-00624-t002:** Properties of base oil.

Flash Point	Closed Cup: ≥220 °C (≥428 °F)
Vapor pressure	≤0.01 kPa (≤0.075006 mm Hg)
Vapor density	≥5 [Air = 1]
Relative density	0.842 [Water = 1]
Partition coefficient: n-octanol/water	3.9 to 6
Auto-ignition temperature	260 to 371 °C (500 to 699.8 °F)
Viscosity	Kinematic (40 °C (104 °F)): 0.36 cm^2^/s (36 cSt)

**Table 3 materials-16-00624-t003:** Statistical analysis results of the rotational viscosity as a function of aged binder contents and the addition of base oil (α = 0.05).

RV	Recycle %	0			50			100		
Recycle %	BO %	0	5	10	0	5	10	0	5	10
0	0	-	S	S	S	N	S	S	S	S
	5		-	S	S	S	N	S	S	S
	10			-	S	S	S	S	S	S
50	0				-	S	S	S	N	S
	5					-	S	S	S	S
	10						-	S	S	S
100	0							-	S	S
	5								-	S
	10									-

BO: Base oil. N: non-significant, S: significant.

**Table 4 materials-16-00624-t004:** Statistical analysis results of the G*/sin *δ* for original condition as a function of aged binder contents and the addition of base oil (α = 0.05).

G*/sin *δ * (Orig.)	Recycle %	0			50			100		
Recycle %	BO %	0	5	10	0	5	10	0	5	10
0	0	-	S	S	S	S	S	S	S	S
	5		-	S	S	S	S	S	S	S
	10			-	S	S	S	S	S	S
50	0				-	S	S	S	S	S
	5					-	S	S	S	S
	10						-	S	S	S
100	0							-	S	S
	5								-	S
	10									-

BO: Base oil. N: non-significant, S: significant.

**Table 5 materials-16-00624-t005:** Statistical analysis results of the G*/sin *δ* for RTFO condition as a function of aged binder contents and the addition of base oil (α = 0.05).

G*/sin *δ * (RTFO)	Recycle %	0			50			100		
Recycle %	BO %	0	5	10	0	5	10	0	5	10
0	0	-	S	S	S	S	S	S	S	S
	5		-	N	S	S	N	S	S	S
	10			-	S	S	S	S	S	S
50	0				-	S	S	S	S	S
	5					-	S	S	S	S
	10						-	S	S	S
100	0							-	S	S
	5								-	S
	10									-

BO: Base oil. N: non-significant, S: significant.

**Table 6 materials-16-00624-t006:** Statistical analysis results of the G*·sin *δ* for RTFO + PAV condition as a function of aged binder contents and the addition of base oil (α = 0.05).

G*·sin *δ* PAV	Recycle %	0			50			100		
Recycle %	BO %	0	5	10	0	5	10	0	5	10
0	0	-	S	S	S	S	S	S	N	S
	5		-	S	S	S	N	S	S	S
	10			-	S	S	S	S	S	S
50	0				-	S	S	S	S	S
	5					-	S	S	S	N
	10						-	S	S	S
100	0							-	S	S
	5								-	S
	10									-

BO: Base oil. N: non-significant, S: significant.

**Table 7 materials-16-00624-t007:** Statistical analysis results of the stiffness and *m-*value for RTFO + PAV as a function of aged binder contents and the addition of base oil at −12 °C (α = 0.05).

Stiffness at −12 °C	Recycle %	0			50			100		
Recycle %	BO %	0	5	10	0	5	10	0	5	10
0	0	-	S	S	N	S	S	S	S	S
	5		-	S	S	N	S	S	S	N
	10			-	S	S	N	S	S	S
50	0				-	S	S	S	S	S
	5					-	S	S	N	S
	10						-	S	S	N
100	0							-	S	S
	5								-	S
	10									-
***m*-value at −12 °C**	**Recycle %**	**0**			**50**			**100**		
**Recycle %**	**BO %**	**0**	**5**	**10**	**0**	**5**	**10**	**0**	**5**	**10**
0	0	-	N	S	S	S	S	S	S	S
	5		-	N	S	S	S	S	S	S
	10			-	N	S	S	S	S	S
50	0				-	S	S	S	S	S
	5					-	N	S	S	S
	10						-	S	S	S
100	0							-	N	N
	5								-	N
	10									-

BO: Base oil. N: non-significant, S: significant.

**Table 8 materials-16-00624-t008:** Statistical analysis results of the stiffness and *m-*value for RTFO + PAV as a function of aged binder contents and the addition of base oil at −18 °C (α = 0.05).

Stiffness at −18 °C	Recycle %	0			50			100		
Recycle %	BO %	0	5	10	0	5	10	0	5	10
0	0	-	S	S	S	S	S	S	S	S
	5		-	S	S	N	S	S	S	N
	10			-	S	S	N	S	S	N
50	0				-	S	S	N	S	S
	5					-	S	S	N	S
	10						-	S	S	S
100	0							-	S	S
	5								-	S
	10									-
***m*-value at −18 °C**	**Recycle %**	**0**			**50**			**100**		
**Recycle %**	**BO %**	**0**	**5**	**10**	**0**	**5**	**10**	**0**	**5**	**10**
0	0	-	S	S	S	N	N	S	S	S
	5		-	N	S	S	S	S	S	S
	10			-	S	S	S	S	S	S
50	0				-	S	S	N	N	N
	5					-	N	S	S	S
	10						-	S	S	S
100	0							-	N	S
	5								-	N
	10									-

BO: Base oil. N: non-significant, S: significant.

**Table 9 materials-16-00624-t009:** Statistical analysis results of the stiffness and *m-*value for RTFO + PAV as a function of aged binder contents and the addition of base oil at −24 °C (α = 0.05).

Stiffness at −24 °C	Recycle %	0			50			100		
Recycle %	BO %	0	5	10	0	5	10	0	5	10
0	0	-	S	S	S	S	S	S	S	S
	5		-	S	S	N	S	S	S	S
	10			-	S	S	N	S	S	S
50	0				-	S	S	N	S	S
	5					-	S	S	S	S
	10						-	S	S	S
100	0							-	S	S
	5								-	S
	10									-
***m*-value at −24 °C**	**Recycle %**	**0**			**50**			**100**		
**Recycle %**	**BO %**	**0**	**5**	**10**	**0**	**5**	**10**	**0**	**5**	**10**
0	0	-	S	S	S	N	S	S	N	N
	5		-	N	S	S	N	S	S	S
	10			-	S	S	N	S	S	S
50	0				-	S	S	N	S	S
	5					-	S	S	N	N
	10						-	S	S	S
100	0							-	S	S
	5								-	S
	10									-

BO: Base oil. N: non-significant, S: significant.

## Data Availability

The data used to support the findings of this study are included within the article.

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
