# Peer review of "Characterization of Base Oil Effects on Aged Asphalt Binders Considering Bicycle Road"

_materials, 2023, doi:10.3390/ma16020624_

Round 1
Reviewer 1 Report
The article is sutable for publication. May I suggest to improve the abstract by including the outcomes.
Author Response
Dear Reviewer,
Thank you for your valuable comment. The paper was revised considering your suggestion.
Thank you.

Reviewer 2 Report
In this contribution, the authors investigated the rejuvenating effect of base oil on aged binders in terms of rotational viscosity, rutting/fatigue cracking properties, and thermal cracking resistance. This research is inspiring to the readership of Materials. I would recommend its publishing if the following questions/comments are addressed.
1. What is the frequency employed for DSR measurements? This frequency simulates the shearing condition corresponding to a traffic speed, e.g., 10 rad/s for 55 mph. Considering the relatively slow speed of bicycles, is it reasonable to test the binders at a lower frequency?
2. What is the difference in softening point and penetration between the original and rejuvenated binders?
3. Does the base oil affect the fire retardancy and aggregate affinity of rejuvenated binders?
4. In Figure 4, PG64-22 (100%) exhibits the comparable rotational viscosity with PG64-22 (50%) + Aged binder (50%) + BO 5%. But the statistical analysis shows a non-significant difference between PG64-22 (100%) and PG64-22 (50%) + Aged binder (50%) + BO 10% instead of 5% (Table 3 and line 160). How to justify the congruency of the experimental result and statistical analysis?
5. The stiff characteristics of the aged binders are attributed to the increasing proportion of asphaltene (Line 177), and the rejuvenation is due to base oil dissolving asphaltene (Line 277). Is the increasing proportion of asphaltene evidenced by experimental results, e.g., SARA analysis, or literature support?
Author Response
Dear Reviewer,
Thank you for your valuable comments.
We answered all your comments and reflected on the paper.
Thank you.

Reviewer 3 Report
Dear Authors
Thanks for your effort, the key objective is to evaluate the result of base oil on the aged binder. Primary, to extract the aged binder, a Rolling thin-film oven (RTFO) and Pressure aging vessel (PAV) were used to emulate the life cycle of asphalt pavement from making to overhaul life, and then individual content of aged binder (0%, 50%, and 100%) was added to PG 64-22. Nice article though, But the following questions and comments should be taken into consideration:
- What is the difference between the load and the environmental conditions of the pavement of the bicycle path compared to other means of transportation?
-What is the reason for choosing bitumen 64-22? While it seems that there is a lot of light and low repetition for the bicycle path.
-Explain why the percentages of selected oil are 5% and...? Mention their references.
Write down the physical and chemical properties of this oil.
-Explain, in Figure 4, the rotational viscosity value is so high for 100% oil?
-It is better to display or draw tables 7 to 9 in a more concise, attractive, and understandable way.
-Explain how to use or mix the base oil in the asphalt mixture.
Thanks
Author Response
Dear reviewer,
Thank you for your valuable comments. We did our best to reflect your comments on the paper.
Please find the attached file.
Thank you.
